# The Role of Receptor–Ligand Interaction in Somatostatin Signaling Pathways: Implications for Neuroendocrine Tumors

**DOI:** 10.3390/cancers16010116

**Published:** 2023-12-25

**Authors:** Agnieszka Milewska-Kranc, Jarosław B. Ćwikła, Agnieszka Kolasinska-Ćwikła

**Affiliations:** 1GENELYTICA Sp. z o.o., Akademicka 48a, 18-400 Łomża, Poland; 2School of Medicine, University of Warmia and Mazury, Aleja Warszawska 30, 10-082 Olsztyn, Poland; 3Diagnostic Therapeutic Center–Gammed, Lelechowska 5, 02-351 Warsaw, Poland; 4“Maria Skłodowska-Curie” National Institute of Oncology, W.K. Roentgena 5, 02-781 Warsaw, Poland; agnieszka.kolasinska-cwikla@pib-nio.pl

**Keywords:** receptor–ligand interactions, neuroendocrine tumors, SSTR characteristics, intracellular signaling, therapeutic targets, tumor development mechanisms, hormone secretion regulation

## Abstract

**Simple Summary:**

The tissue expression of somatostatin receptor (SSTR) subtypes in neuroendocrine tumors (NETs) holds paramount significance with far-reaching diagnostic and therapeutic implications. SSTRs contribute significantly to localizing NETs, aiding in advanced imaging techniques such as somatostatin receptor scintigraphy and PET scans. Elevated SSTR expression often aligns with less aggressive tumors, impacting tumor grading and influencing treatment decisions. Therapeutically, SSTRs serve as pivotal targets for treatments like targeted radiotherapy and synthetic somatostatin analogs (e.g., octreotide), commonly employed to inhibit hormone secretion and suppress tumor growth. Tailoring treatment strategies based on SSTR subtype expression enhances therapeutic efficacy. SSTR expression serves as a valuable prognostic indicator, predicting treatment response. While challenges like tumor heterogeneity persist, ongoing research explores promising avenues, including combination therapies. In conclusion, a nuanced understanding of SSTR expression in NETs is indispensable for precise diagnosis and the development of tailored therapeutic interventions, offering the potential for improved patient outcomes.

**Abstract:**

Neuroendocrine tumors (NETs) arise from neuroendocrine cells and manifest in diverse organs. Key players in their regulation are somatostatin and its receptors (SSTR1–SSTR5). Understanding receptor–ligand interactions and signaling pathways is vital for elucidating their role in tumor development and therapeutic potential. This review highlights SSTR characteristics, localization, and expression in tissues, impacting physiological functions. Mechanisms of somatostatin and synthetic analogue binding to SSTRs, their selectivity, and their affinity were analyzed. Upon activation, somatostatin initiates intricate intracellular signaling, involving cAMP, PLC, and MAP kinases and influencing growth, differentiation, survival, and hormone secretion in NETs. This review explores SSTR expression in different tumor types, examining receptor activation effects on cancer cells. SSTRs’ significance as therapeutic targets is discussed. Additionally, somatostatin and analogues’ role in hormone secretion regulation, tumor growth, and survival is emphasized, presenting relevant therapeutic examples. In conclusion, this review advances the knowledge of receptor–ligand interactions and signaling pathways in somatostatin receptors, with potential for improved neuroendocrine tumor treatments.

## 1. Somatostatin Receptors: Localization and Function

Somatostatin receptors, protein structures present on the surface of cells, play a key role in the regulation of various physiological processes. They are stimulated by somatostatin, a peptide hormone that affects the release of other hormones and regulates the functions of the digestive, endocrine, and nervous systems. In addition, the somatostatin receptor can also bind to other ligands, which opens the possibility of exploring other signaling pathways and therapeutic applications in the field of diagnosis and treatment, especially in diseases such as neuroendocrine tumors. Five major subtypes of somatostatin receptors have been described in the human body, known as SSTR1-SSTR5 [1]. Each of these receptors differs in its structure (Table 1), expression in different tissues, and biological function (Table 2).

The structure of genes encoding somatostatin receptors and receptor proteins differs in terms of organization and sequence of nucleotides and amino acids. Genes encoding particular receptor subtypes are located on different chromosomes [2] (Table 1). In their promoter regions, there are sites that undergo epigenetic modifications and control the expression of this gene through interactions with transcription factors [3]. The expression profile of this gene is also affected by the 5′ and 3′ UTR regions, which contain regulatory sequences affecting the translation process and mRNA stability [1,4,5].

The structure of the SSTR protein includes seven transmembrane α-helical domains (TM1-TM7), connected by cytoplasmic (ICL1-ICL3) and extracellular (ECL1-ECL3) fragments [1,5]. Each domain has its own unique function and can interact with other molecules inside the cell or on its surface. Transmembrane alpha-helices form a channel for somatostatin signals, passing through the cell membrane [6,7]. Between the transmembrane helices are extracellular and intracellular loops. Extracellular loops exposed to the outside of the cell contain ligand binding sites, allowing the somatostatin receptor to bind to this peptide. Intracellular loops are found inside the cell and can interact with signaling proteins such as G proteins, protein kinases, and other components of signaling pathways [6,7,8,9]. In addition, somatostatin receptor proteins have conservative motifs characteristic of G-protein-coupled receptors, such as the DRY motif, located in the third inner loop and responsible for signal activation [10,11].

**Table 1 cancers-16-00116-t001:** Comparison of molecular structure, genes, and proteins of somatostatin receptors (SSTR1-5) and their genetic and epigenetic modifications, and regulatory factors (based on [1,2,5,6,12,13,14,15,16]).

Gene	SSTR1	SSTR2	SSTR3	SSTR4	SSTR5
**Genomic location** (by HGNC)	Chromosome 14: 14q13	Chromosome 17: 17q25.1	Chromosome 22: 22q13.1	Chromosome 20: 20p11.21	Chromosome 16: 16p13.3
**Exon count**	3	2	6	1	2
**Transcriptional Factors**	e.g., POLR2A, RAD21, STAT3, HNF4A, ZNF48, SP1, NCOR1, CREB1, FOS	e.g., AML1a, AP-1, En-1, ER-alpha, GATA-2, MyoD, NF-kappaB, Pbx1a, ZIC2	e.g., SP1, ZNF207, JUND, TCF7, GATA2, SMAD5, EGR2, RAD51, WT1	e.g., HNF4A, FOXJ2, Meis-1, NF-kappaB, p53, Pax-4a	e.g., Sp1, Bach2, C/EBPalpha, CREB E47, LCR-F1, p53, Pax-4a
**Epigenetic modifications affecting expression**	Hypermethylation promoter	Differential methylation of promoter regions, histone modifications (e.g., H3K4 methylation), ubiquitination	Promoter methylation	Hypomethylation of the promoter region	Methylation of the 4 CpG islands
**MicroRNA regulators**	e.g., miR-4285, miR-149-3p, miR-1207-5p, miR-502-3p	e.g., miR-378a-5p, miR-191-3p, miR-106a-3p, miR-122-5p, miR-130b-3p, hsa-miR-203b-3p, miR-215-5p	e.g., miR-328-5p, miR-623, miR-3667-3p, miR-766-3p	miR-132, miR-211, miR-204	miR-129-5p, miR-204, miR-375
**Sequence variations with clinical significance**	V351A, R380P, L326V, V153L (missense variants), rs146928458 (synonymous variant)	I87V, T30N, N22S, N318S (missense variants)	S7T, E288K, I125T, V175L, V145M (missense variants)	F114L, R377H, V125D, C366Y, S253W (missense variants)	R137H, P17R, R154H, R248H, P17L (missense variants)
**Structural variations**	nsv832773 (loss), esv3584282 (gain), nsv952830 (deletion)	nsv833534 (loss), nsv4263666 (deletion)	nsv4278013 and nsv966108 (duplications); nsv829203, nsv459883, and nsv834188 (loss)	nsv4336704 (sequence alteration), nsv4276734 (duplication), nsv524981 and nsv520810 (loss)	esv2422427 (duplication), nsv952898 (deletion), nsv457315 and nsv571046 (gain), sv471066 (loss)
**Differences in protein sequence**	391 amino acids; longer extracellular fragments in domains 1, 2, 4, and 7	369 amino acids; shorter extracellular fragments in domains 2, 4, and 7	418 amino acids; longer extracellular fragments in domains 1, 2, and 7	388 amino acids; shorter extracellular fragment in domain 3	364 amino acids; longer extracellular fragments in domains 1, 3, and 4

In the superfamily of G-protein-coupled receptors (GPCRs), there are mechanisms that generate splicing variants that have fewer than seven transmembrane domains [17,18]. These truncated receptors, such as the long and short isoforms of the SSTR2 gene (SSTR2A and SSTR2B), may have unique functions and be associated with neoplastic pathology [17,18,19,20]. Recent studies have shown the existence of two functional truncated isoforms of human SSTR5 subtypes (sst5TM5 and sst5TM4) with different expression patterns in normal tissues and different types of pituitary tumors [21]. These isoforms show different functional responses to somatostatin.

The localization and expression of somatostatin receptors are important for understanding the role of these receptors in various tissues and organs. Studies have shown that individual SSTR subtypes show different expression patterns. In addition, they may vary depending on physiological and pathological conditions, as well as depending on the specificity of a given organism [1,22,23]. The SSTR1 receptor is widely expressed in various tissues, including the nervous system, pancreas, salivary glands, basal ganglia, heart, and blood vessels. In the brain, SSTR1 is highly distributed in the hippocampus, cerebral cortex, hypothalamus, and lateral nucleus accumbens [1,6,24,25]. In the pancreas, the presence of SSTR1 receptors has been observed in α-cells secreting glucagon and δ-cells secreting somatostatin [26,27]. The presence of SSTR1 receptors in the heart and blood vessels suggests that somatostatin may be involved in the regulation of cardiac and vasomotor functions [28].

The SSTR2 receptor is present in many tissues, such as the nervous system, pancreas, gastrointestinal tract, liver, adrenal glands, and thyroid gland [6,24]. In the brain, the SSTR2 receptor is present in the hippocampus, cerebral cortex, hypothalamus, thalamus, lateral nucleus accumbens, and posterior brain [25,29]. In the pancreas, SSTR2 receptors have been detected in both α and β cells [26,27,30]. SSTR2 receptors are also detected in the gastrointestinal tract, liver, adrenal glands, and thyroid gland [31,32].

The SSTR3 receptor is expressed in the nervous system, gastrointestinal tract, pancreas, salivary glands, and basal ganglia [1,24]. In the brain, the SSTR3 receptor is present in the hippocampus, cerebral cortex, hypothalamus, thalamus, lateral accumbens, and amygdala [25]. In the pancreas, the presence of SSTR3 receptors has been found mainly in δ cells [27]. In addition, SSTR1-3 receptors are detected in the parotid, sublingual, and submandibular glands [33,34].

The SSTR4 receptor Is expressed in the brain, pancreas, and in some types of cancer [1,24]. In the brain, the SSTR4 receptor is present in the hippocampus, cerebral cortex, hypothalamus, and lateral nucleus accumbens [35,36]. In the pancreas, the presence of SSTR4 receptors has been found primarily in α cells [27].

The SSTR5 receptor is expressed in various tissues, including the nervous system, pancreas, basal ganglia, gastrointestinal tract, thyroid, and adrenal glands [1,24,37]. In the brain, the SSTR5 receptor is present in the hippocampus, cerebral cortex, hypothalamus, thalamus, lateral accumbens, and amygdala [38,39]. In the pancreas, SSTR5 receptors have been detected mainly in α-cells, although they may also be expressed in other cell types [26,27]. The expression of SSTR2 and SSTR5 receptors in the thyroid gland suggests the possibility of thyrotropin secretion being regulated by somatostatin [40].

**Table 2 cancers-16-00116-t002:** Physiological and molecular characteristics of somatostatin receptor subtypes and their analogues (based on [41,42,43,44,45,46,47,48,49,50,51,52,53,54,55]).

*Subtype Receptor*	*Tissue Localization*	*Intracellular Signalling*	*Physiological Functions*
*Somatostatin receptor type 1*	Nervous system, stomach, colon, small intestine, lung, liver, kidney, duodenum, pancreas, salivary glands, basal ganglia, heart, blood vessels	Inhibition of cAMP, inhibition of cGMP production	Regulation of hormone secretion (e.g., insulin, glucagon), impact on cognitive processes, neuroprotective effects
*Somatostatin receptor type 2* splicing variants A and B	Nervous system, kidney, spleen, stomach, appendix, liver, adrenal glands, thyroid	Inhibition of cAMP, activation of phospholipase C (PLC) pathway	Inhibition of growth hormone, insulin, glucagon secretion, regulation of gastrointestinal motility, inhibition of bile secretion
*Somatostatin receptor type 3*	Nervous system, testis, ovary, gastrointestinal tract, pancreas, salivary glands, basal ganglia, lymph node	Inhibition of cAMP, activation of phospholipase C (PLC) pathway	Regulation of hormone secretion (e.g., insulin, glucagon), impact on cognitive processes, regulation of gastrointestinal motility
*Somatostatin receptor type 4*	Nervous system, gastrointestinal tract, pancreas, salivary glands, heart, blood vessels	Inhibition of cAMP	Regulation of hormone secretion (e.g., insulin, glucagon), regulation of gastrointestinal motility, impact on heart and blood vessel function
*Somatostatin receptor type 5*	Nervous system, kidney, adrenal, gallbladder, stomach, prostate, duodenum, colon	Inhibition of cAMP, activation of phospholipase C (PLC) pathway, activation of MAP kinase pathway	Regulation of hormone secretion (e.g., insulin, glucagon, thyrotropin, cortisol), impact on cognitive processes, regulation of gastrointestinal motility, anti-proliferative effects

Somatostatin receptors, which are found in various tissues of the body, play an important role in the regulation of their physiological processes. One of the most important effects of somatostatin receptors is the inhibition of growth hormone secretion by the anterior pituitary gland. In this process, the SSTR2 and SSTR5 receptors play a key role [52,56,57].

In addition, the same somatostatin receptor subtypes are involved in the regulation of insulin secretion from pancreatic beta cells and glucagon from pancreatic alpha cells [30,58]. As a result, somatostatin and its receptors play an important role in maintaining the body’s glycemic balance. SSTRs affect the secretion of gastrin, a hormone responsible for regulating stomach function. SSTR2 and SSTR5 receptors are responsible for the inhibition of gastrin secretion from G cells of the gastric mucosa [59,60]. This regulation affects digestive processes and gastrointestinal metabolism. SSTRs are also involved in regulating gut motility. SSTR1 and SSTR2 receptors play a role in inhibiting peristalsis and intestinal secretion [61,62,63]. Through these mechanisms, somatostatin regulates intestinal function and accelerates the absorption of nutrients.

Somatostatin receptors also affect the functions of the nervous system. SSTR1, SSTR2, and SSTR4 receptors are present in neurons and modulate neuronal transmission by affecting the release of neurotransmitters. For example, activation of SSTR2 receptors inhibits glutamate release, which may affect synaptic transmission. Stimulation of SSTR5 receptors may affect the perception of pain by modulating the transmission of pain signals in the brain. SSTR2 and SSTR4 receptors play a role in protecting neurons by inhibiting apoptosis and the inflammatory immune response. Activation of these receptors can protect neurons from damage and death. In turn, SSTR2, when activated, may affect sleep control, including sleep induction and prolongation of REM sleep [64,65,66,67].

Modulation of the expression of somatostatin receptors is important for the regulation of their function in various tissues and pathological conditions. Many mechanisms of mechanisms that can affect the expression of SSTR receptors have been described, and the expression of genes encoding SSTR receptors can be regulated at the transcriptional level by various factors [68] (Table 1). For example, the transcription factor CREB (cAMP response-element-binding protein) can influence the expression of the SSTR1, SSTR2, and SSTR5 genes [68,69]. Another example is the transcription factor STAT3 (signal transducer and activator of transcription 3), which can modulate the expression of SSTR2 and SSTR3 [27]. External factors such as hormones, cytokines, chemicals, or growth factors can affect the regulation of SSTR receptor expression. For example, somatostatin can induce the expression of its receptors through positive feedback, creating an autoregulatory mechanism [1]. Stimulation of the β-adrenergic receptor may also increase the expression of SSTR2 receptors in some tissues [70]. In addition, changes in chromatin structure and DNA methylation may affect the expression of SSTR genes. For example, methylation of the SSTR1, SSTR2, and SSTR5 gene promoters may influence their expression in cancer cells [14,37]. The use of histone deacetylase inhibitors may also affect the expression of SSTR receptors [71].

## 2. Diagnostics of the Expression of Somatostatin Receptors

Imaging diagnosis of SSTR receptors in neuroendocrine tumors plays a key role in the identification, localization, and evaluation of these tumors. In recent years, the development of new perspectives and research methods has brought significant progress in the diagnosis of the expression of these receptors. The variety of methods for identifying and localizing somatostatin receptors in neuroendocrine tumors includes both in vivo and in vitro approaches, providing in-depth analysis of the expression of these receptors at the cellular level. The ability to accurately determine the presence and distribution of SSTR receptors allows physicians to select appropriate therapeutic strategies and monitor therapy progress. There are several advanced diagnostic techniques used to detect SSTR expression in neuroendocrine tumors. One of the primary methods is immunohistochemistry, which involves the use of SSTR-specific antibodies in tissue samples. This method allows for the identification of specific types of receptors [72,73]. Molecular tests such as PCR can be used to detect the expression of SSTR genes [74,75]. These tests provide information on the level of SSTR gene expression in the tumor, which may be of significant prognostic and therapeutic importance. However, it should be remembered that the presence of mRNA is not always synonymous with the expression of receptor proteins at the cellular level. One of the very popular in vivo methods is receptor/ocreotide scintigraphy. In this method, patients are administered a radiolabeled somatostatin analogue-octreotide that has the ability to bind to receptors. A special gamma scanner is then used to record the emission of radiation [22,53,76,77,78,79]. Scintigraphic imaging allows for the precise localization of the tumor and assessment of its extent. Positron emission tomography (PET) is another advanced molecular imaging method that uses radiolabeled somatostatin analogs. The most commonly used labeled analogs include ^111Indium-pentetreotide (^111In-pentetreotide) and ^68Gallium-DOTA-TATE (^68Ga-DOTA-TATE). After administration of these labeled analogues, their molecules bind to areas expressing SSTR, which allows for the detection of radiation sources in the body and precise localization of NETs, especially those with high SSTR2 expression. After administering a labeled somatostatin analogue to the patient, PET records the emission of positrons generated during the decay of the radioactive isotope [80,81,82]. PET images provide detailed information on the presence of NETs that show high accumulation of radiolabeled somatostatin analogue.

Detection of SSTR expression in neuroendocrine tumors is important for individualization of treatment and therapeutic decisions. Identifying the presence and distribution of receptors allows physicians to accurately plan treatments, such as radiolabel therapy, which uses radioactive tracers attached to somatostatin analogues to selectively destroy SSTR-expressing cancer cells.

## 3. Ligand–Receptor Interactions in Neuroendocrine Tumors

Ligand–receptor interactions play a key role in numerous biological processes, including in the context of neuroendocrine tumors. Neuroendocrine neoplasms (NEN) are currently relatively rare and heterogeneous group of neoplasms arising from dispersed cells found in the respiratory system, thymus, gastrointestinal system, and also uncommon neural crest-derived endocrine cells or organs known as paraganglia. They are characterized by the ability to produce and secrete hormones and bioactive peptides that affect various aspects of the body’s functioning [83,84].

One of the important aspects of the pathogenesis of neuroendocrine tumors (NET), which are built with well or moderate differentiated tumor cells of NEN, is the interaction between ligands, such as somatostatin and its analogues, and somatostatin receptors. Native somatostatin is a 14 amino acid cyclic peptide hormone that acts as an endogenous SSTR ligand. By interacting with somatostatin receptors present on the surface of tumor cells, somatostatin affects the processes of growth, differentiation, and hormonal secretion of these tumors [85,86,87]. The discovery of somatostatin and its role in hormonal regulation has increased our knowledge of neuroendocrine tumors [88]. In addition, it is important to understand the role of somatostatin analogues, which are structurally similar to somatostatin but have chemical modifications that improve their pharmacokinetics and pharmacodynamics. Somatostatin analogues, such as octreotide, lanreotide, or pasireotide, show increased selectivity and affinity to somatostatin receptors (Table 3), which enables their use in the treatment of neuroendocrine tumors [41,89,90].

Binding of somatostatin and its analogues to SSTR is a complex process that involves molecular interactions between the ligand and the receptor at the structural level. These interactions are mainly determined by key sites in the structure of these ligands and in the receptors. Structural studies have shown that amino acids in the N-terminal region of somatostatin, such as cubic proline and aspartic acid, play an important role in establishing contacts with SSTR receptors [1,91,92,93]. In the case of somatostatin analogues, such as octreotide, chemical modifications that may affect binding to receptors are important. Crystallographic studies of SSTR receptors revealed the presence of a special ligand-binding pocket inside the receptors. In these pockets, key fragments of the ligands come into contact, allowing for stable binding [94,95,96].

Somatostatin and its analogues bind to SSTR receptors by interacting with several receptor segments. For example, somatostatin binding to the SSTR1 and SSTR4 receptors involves amino acid fragments in the transmembrane region of the receptors, including the TM2 and TM3 segments [94,97]. For the SSTR2 receptor, the TM6 and TM7 segments play an important role in octreotide binding [98]. Individual somatostatin analogues may show differences in affinities for individual subtypes of SSTR receptors. For example, octreotide has a high affinity for the SSTR2 and SSTR5 receptors, while pasireotide has the greatest affinity for the SSTR5 receptors (Table 3). These differences result from differences in the structure of ligands and in the structure and expression of receptors (Table 1). Upon binding of somatostatin or its analogue to the SSTR, a conformational change of the receptors occurs, which leads to the activation of intracellular signaling pathways. The detailed mechanisms of signal initiation for individual receptor subtypes may be different.

After somatostatin binds to its receptors, a series of cellular events occur [99]. The receptor activates the G protein, which is heterotrimeric and consists of α, β, and δ subunits. Activated G protein depends on the specific SSTR subtype and can be one of three Gαi protein isoforms (Gαi1-3). There are also reports that SSTR2 may interact with the Gαo2/β2/γ3 complex in pituitary cells, controlling the activity of calcium channels [100,101], and SSTR3 may interact with Gαo, Gα14, and α16 [102]. Upon activation of the G protein, various intracellular secondary messenger systems are modulated. Activation of different SSTR subtypes leads to the activation of specific signaling pathways.

An important role in GPCR signaling after ligand binding is played by GPCR-interacting proteins (GIPs), which regulate G protein signaling (RGS) and GPCR kinases (GRKs) [103,104,105,106]. GIP proteins can affect receptor binding or change their functional responses, creating many potential receptor–protein interactions [103]. In turn, RGS proteins regulate GPCR responses by stimulating Gα protein GTPase activity activated by the receptor [104,105,106]. The phosphorylation of SSTR receptors by GRK is another significant event in the signaling process. The GRK family consists of six serine/threonine kinases that bind to and phosphorylate activated GPCR agonists [107,108,109]. The phosphorylation of the receptors leads to the recruitment of cytoplasmic proteins called arrestins. Arrestins play a key role in GPCR densitization by blocking further interactions between receptors and G proteins, leading to signal quenching. Furthermore, arrestins are involved in receptor endocytosis by directing GPCRs to clathrin-coated vesicles [110,111,112,113,114]. Different SSTR subtypes have different capacities for internalization upon agonist binding. For example, SSTR2, SSTR3, and SSTR5 are internalized to a greater extent than SSTR1 and SSTR4 [64,115,116,117]. Once internalized, SSTR3 and SSTR5 are further degraded in lysosomes, while SSTR2 is rapidly recirculated to the cell surface [113,118]. Also, heterodimerization of SSTR receptors is essential for their function. SSTR receptors can form homodimers or heterodimers with other SSTR subtypes or other types of GPCRs. Heterodimerization can affect receptor properties such as ligand affinity, signal transduction, densitization, and internalization [119,120,121,122]. SST ligand binding to SSTR receptors leads to the activation of various intracellular signaling pathways, and the exact mechanisms by which SSTRs regulate signaling are still being intensively studied.

## 4. Intracellular Signaling Pathways of Somatostatin Receptors

Following the receptors by somatostatin, there is activation of various signaling pathways that affect cell function [10,52,123]. The five receptors exhibit common signaling pathways, including the inhibition of adenylate cyclase, activation of phosphotyrosine phosphatase, and modulation of mitogen-activated protein kinase (MAPK) through G-protein-dependent mechanisms [1]. These shared pathways contribute to the coordinated regulation of cellular processes mediated by these receptors.

One of the key signaling pathways associated with somatostatin receptors is the cAMP (cyclic adenosine monophosphate) pathway. After activation of somatostatin receptors, cAMP production is inhibited by the G-protein complex. A decrease in cAMP levels leads to different effects depending on the type of cell, such as inhibition of hormone secretion, reduction of cell proliferation, and regulation of ion channels [52,124].

Another important signaling pathway is the phospholipase C (PLC) pathway. Upon activation of the somatostatin receptors, G-proteins activate the PLC, which catalyzes the breakdown of phosphatidylinositol bisphosphate (PIP2) into inositol triphosphate (IP3) and diacylglycerol (DAG). IP3 leads to the release of calcium ions from intracellular stores, which initiates further signals in the cell. DAG activates kinase C proteins that play a role in the regulation of many cellular processes.

MAP kinases (mitogen-activated protein kinases) are another key mediator of signaling after activation of somatostatin receptors [125,126,127]. Depending on the cell type, activation of somatostatin receptors can lead to activation of the MAP kinase pathway, such as ERK (extracellular-signal-regulated kinase), JNK (c-Jun N-terminal kinase), and p38 MAP kinase [56,108,109,110,127,128]. These signaling pathways can influence various cellular processes including proliferation, differentiation, apoptosis, and metabolism.

In addition to these signaling pathways, activation of somatostatin receptors may also affect other pathways, such as the PI3K/Akt (phosphatidylinositol 3-kinase/Akt) pathway, the mTOR (mammalian target of rapamycin) pathway, and the NF-κ B (nuclear factor kappa-light) pathway (chain-enhancer of activated B cells), which play important roles in the regulation of cellular processes [129,130].

The activity of SST and its analogues is also contingent upon specific binding to cell- and tissue-specific receptors, with the mode of action being context dependent. Within the central nervous system, SST operates as a neurotransmitter, facilitating communication between neurons. Additionally, its hypothalamic–hypophyseal transfer qualifies it as a neurohormone, participating in neuroendocrine regulation [131,132]. Beyond the confines of the central nervous system, SST assumes a dual role, acting as a paracrine factor to regulate neighboring cells and an autocrine factor to facilitate self-regulatory processes [133,134]. Moreover, upon secretion into the intestinal lumen, it manifests as a ‘lumone’, exerting direct effects on gut physiology [135].

## 5. Role of Somatostatin Receptors in Neuroendocrine Tumors

Neuroendocrine tumors are a diverse group of neoplasms originating from various endocrine glands, including the pituitary, parathyroid, and neuroendocrine adrenal glands, as well as from endocrine islets within the thyroid or pancreas. Additionally, they can develop from endocrine cells interspersed among exocrine cells throughout the digestive and respiratory tracts [136,137,138,139,140,141]. They are characterized by slow growth and variable clinical prognosis [139,140]. NETs can be categorized based on proliferative activity using the mitotic index and the Ki-67 index. G1 NETs have a low mitoses number and low Ki-67 index, G2 NETs have moderate proliferative activity, and G3 NETs are poorly differentiated with a high mitoses number or Ki-67 index [142,143,144,145]. NETs can be functional or non-functional depending on their ability to secrete specific hormones [80,83]. For example, pituitary tumors (PitNETs) may secrete GH, PRL, or other pituitary hormones, causing endocrine disruption and characteristic clinical symptoms [146,147,148,149,150]. Pancreatic neuroendocrine tumors are often inactive and do not cause significant symptoms, but some functioning PanNETs can secrete various hormones such as insulin, gastrin, ghrelin, VIP, glucagon, and SST [58,151]. GEP NETs (gastroenteropancreatic cancer tumors) are usually asymptomatic, but functioning tumors may produce bioactive peptides or amines, such as histamine in gastric NETs or secretin, gastrin, and motilin in the duodenum [141,152,153,154]. NETs can also modify secreted hormones and peptides at the genetic level, leading to different variants of these substances, for example, different forms of gastrin [155]. Non-functional NETs are usually diagnosed at a later stage, after the onset of symptoms related to tumor mass or metastases [146,147].

SSTRs are found on the surface of tumor cells and have the potential to serve as diagnostic markers and as therapeutic targets for somatostatin analogues (SSAs). Somatostatin receptors are expressed in a substantial proportion of these tumors, ranging from 80% to 90% of cases, except tumor insulin (insulinoma), where SSTR expression applies only 50% of patients [156,157,158]. SSTR expression varies by tumor type, but most NETs express all receptor subtypes [159,160,161]. SSTR2A and SSTR5 are most often expressed, while SSTR4 is expressed at a lower level or not detected at all in NETs [159,160,161,162]. SSTR expression is an important prognostic factor and predictor of response to SSA therapy. NETs with higher SSTR expression have been experimentally shown to have good differentiation and are usually classified as G1 or G2 tumors [145,162,163,164,165,166]. Higher expression of SSTR receptors, especially SSTR2A, is often associated with a better prognosis in patients [159,161,163,164]. However, research findings regarding the role of SSTR and SST signaling in NETs depend on tumor type as well as on other tumor-specific factors.

Somatostatin signaling via SSTR receptors plays an important role in the regulation of many tumor cell processes (Figure 1). SST (or its synthetic analogues) inhibits adenyl cyclase activity, which leads to a decrease in the level of the second messenger-cyclic AMP (cAMP) and calcium in the cells, resulting in a decrease in hormone secretion [167,168]. In addition, SST signaling regulates the cell cycle by activating the tyrosine proteins of the phosphatases SHP-1 and SHP-2, which leads to reduced cell proliferation by upregulating cell cycle inhibitors such as p27 and p21 and inhibiting the PI3K/AKT and MAPK signaling pathways [168,169,170]. Stimulation of SSTR2 receptors not only inhibits the release of hormones by the tumor but also inhibits the growth of cancer cells. Additionally, SST2 and SST3 receptors play a role in inducing apoptosis in neuroendocrine tumor cells [162,167]. The intravascular presence of SST suggests that somatostatin and its synthetic analogues may act as anti -angiogenesis agents. Signaling through SST2 leads to inhibition of hormone release and inhibition of tumor cell proliferation, while stimulation of SSTR2 and SSTR3 induces apoptosis [41,171,172,173,174,175]. Despite these intriguing findings, the precise mechanistic underpinnings of SST and its analogues in diverse cellular contexts remain a subject of ongoing investigation. The exploration of these molecular pathways holds the promise of advancing therapeutic approaches for NETs, leading to the development of more targeted and effective treatment strategies for patients in need [176,177].

## 6. Somatostatin and Its Analogues in Neuroendocrine Tumor Therapy

The expression of somatostatin receptors in neuroendocrine tumors plays a crucial role in their imaging recognition and significantly influences the treatment process, especially in the context of metastases. In the pre-treatment phase, high levels of SSTR expression in NETs are a significant target for imaging diagnostic techniques such as somatostatin receptor scintigraphy (SRS) or positron emission tomography (PET). These methods enable the precise localization of tumors and metastases, which is crucial for developing a therapeutic strategy [178,179]. Additionally, the detection and quantitative assessment of SSTR expression can provide prognostic information and aid in assessing the response to treatment. During treatment, especially in the context of peptide receptor radionuclide therapy (PRRT) using radiolabeled somatostatin analogs, SSTR expression is pivotal. Radioactive somatostatin ligands, such as lutetium-177-DOTATATE, bind to SSTR on the surface of tumor cells, allowing a targeted delivery of radiation to the tumor. SSTR expression becomes a key predictive biomarker for the effectiveness of therapy [180,181,182]. Monitoring changes in SSTR expression during treatment can assist in adjusting the therapeutic strategy and evaluating the response to therapy.

Due to the short half-life (less than 3 min), the therapeutic use of somatostatin is difficult. This problem is solved by synthetic analogues that are more resistant to proteolytic enzymes. Somatostatin analogues (SSAs) are revolutionizing the diagnosis and treatment of neuroendocrine tumors [51,171,175,183,184,185]. Octreotide was the first biostable somatostatin analogue with strong affinity for SSTR2- and SSTR5-type receptors (Table 3). Octreotide is used subcutaneously or intravenously in several daily injections. Octreotide’s affinity for somatostatin receptors allows it to inhibit hormone release and tumor cell growth [90,163,186,187,188,189,190]. Another available somatostatin analogue is lanreotide, which has similar effects to octreotide (Table 3). Lanreotide has affinity for SSTR2, SSTR3, and SSTR5 receptors. It is used every 28–56 days in deep subcutaneous or intramuscular injections. It inhibits hormone release and tumor cell growth [191,192,193]. Pasireotide, a second-generation somatostatin analogue, is a ligand for SSTR1, SSTR2, SSTR3, and SSTR5 receptors, which increases their therapeutic potential. Pasireotide is available as a subcutaneous formulation that requires twice-daily administration, as well as a long-acting (LAR) formulation that allows monthly intramuscular administration. Pasireotide’s affinity for various somatostatin receptors enables it to effectively control symptoms and inhibit tumor growth [194,195,196,197,198,199,200]. Research into improving SSA therapy continues. Investigated treatment modalities include oral non-peptide somatostatin type 2 receptor agonists and new drugs such as somatoprim, with affinity for SSTR2, SSTR4, and SSTR5 receptors [201,202]. These innovative therapeutic approaches may contribute to more effective treatment of patients with NETs. SSAs provide new treatment options for neuroendocrine tumors, and further research aims to develop more effective therapies that will be based on more selective affinity for different somatostatin receptors.

Somatostatin analogues show several advantages compared to endogenous somatostatin. First, they have been optimized to maintain their therapeutic effect over a longer period of time, which means less frequent dosing and greater patient comfort. Secondly, somatostatin analogues show greater selectivity. They are designed to have greater affinity for specific somatostatin receptors, such as SSTR2 and SSTR5. These receptors are responsible for suppressing hormone secretion and tumor growth. Owing to that, somatostatin analogues can more effectively inhibit excessive hormone production in neuroendocrine tumors and limit the growth of cancer cells. Third, somatostatin analogues are more biologically stable. All these features make somatostatin analogues effective tools in the treatment of neuroendocrine tumors. Due to their longer duration of action, selectivity, and stability, they provide effective control of clinical symptoms, suppression of excessive hormone secretion, and reduction of tumor growth.

Studies on the anti-proliferative effects of somatostatin analogues have shown that the anti-cancer effect of these drugs may be the result of direct and indirect mechanisms. In the case of direct mechanisms, the activation of somatostatin receptors on tumor cells leads to the inhibition of signal transduction pathways inside the cells. In vitro studies, using cell lines transfected with somatostatin receptors, have shown that all receptor subtypes can inhibit cell proliferation, while specific receptor subtypes (SST2, 3) can mediate apoptosis. These actions are mainly regulated by the MAP kinase signaling pathway and the activation of phosphotyrosine phosphatases [52,56,203]. Among the indirect mechanisms of the anticancer effect of SSAs is the inhibition of the action of mitogenic growth factors, such as insulin-like growth factor (IGF) [204]. Moreover, the inhibition of tumor angiogenesis plays an important role in the anti-proliferative effect SSAs by interacting with somatostatin receptors on endothelial cells and monocytes. The antiproliferative effect may also be related to the activation of phosphotyrosine phosphatases, such as SHP-1 and SHP-2, which affect signaling pathways that regulate cell growth (Figure 1). For example, activation of SSTR2 can inhibit PI3 kinase activity and lead to cell growth arrest by stimulating SHP-1 [205,206]. Clinical studies, such as the PROMID study, have provided evidence of the anti-proliferative effect of octreotide in patients with neuroendocrine tumors of the gastrointestinal tract. In the PROMID study, patients with neuroendocrine tumors who received octreotide LAR had a significantly longer time to disease progression compared to the placebo group. This effect was observed regardless of tumor functionality, chromogranin A level, or patient age. Additionally, patients with lower tumor burden showed the greatest disease stabilization under the influence of octreotide LAR [90]. The CLARINET study confirmed that lanreotide Autogel^®^ has a significant antiproliferative effect in patients with inactive neuroendocrine tumors of the gastrointestinal tract and pancreas. Treatment with lanreotide Autogel^®^ resulted in a significant reduction in the risk of relative disease progression by 53% compared to the placebo group. This effect was observed regardless of the tumor stage and the level of the Ki-67 proliferation index [207]. Similarly, a study by Faniello et al. focused on the role of somatostatin analogues in small intestinal NETs. They demonstrated that pasireotide stimulates apoptosis and inhibits NET cell proliferation by activating the p38 MAPK signaling pathway. SST1 receptors played a key role in mediating these effects [56]. It is also worth noting that studies have shown a synergistic potential in the treatment of NETs when somatostatin analogues are used in combination with other targeted therapies. For example, the combination of SSA with mTOR inhibitors, such as everolimus, was shown to be more effective in inhibiting NET cell growth in a study by Yao et al. [208].

Scientific and clinical studies clearly indicate the positive effect of SSAs on the apoptosis process. On pancreatic cancer cells, it showed that octreotide, one of the popular somatostatin analogues, activates intrinsic apoptosis pathways via SSTR2 and SSTR3. Stimulation of these receptors causes the activation of caspases enzymes, which lead to the degradation of anti-apoptotic proteins, which ultimately contributes to the death of cancer cells [209,210]. In turn, in a study focusing on pasireotide (SOM230), it was shown that it can affect angiogenesis processes and inhibit cell proliferation in pituitary adenomas secreting adrenocorticotropic hormone (ACTH). These effects may be related to the activation of apoptosis by pasireotide via SSTR1, SSTR2, SSTR3, and SSTR5 [211]. Guillermet et al. [212] showed that stable transfection of SSTR2 into BxPC-3 cells, which previously did not express this receptor, resulted in a significant increase in apoptosis. SSTR2 expression increased the sensitivity of these cells to various forms of ligand-induced apoptosis, such as TNF-alpha, TRAIL, and CD95L. The SSTR2-dependent mechanism involved the activation of execution caspases, key factors in apoptosis, and the regulation of the expression of receptors and proteins associated with apoptosis. These studies suggest that gene therapy using both SSTR2 and necrosis ligands may be a potential treatment approach for chemotherapy-resistant pancreatic adenocarcinomas.

Somatostatin analogues also have a significant effect on angiogenesis in neuroendocrine tumors [136,175]. The direct effect of SSAs on angiogenesis results from the activation of type 2 [213]. Activation of SSTR2 inhibits the migration and proliferation of these cells, which reduces the formation of new vessels and reduces tumor growth. In addition, SSAs reduce the concentration of proangiogenic growth factors, such as VEGF (vascular endothelial growth factor), which additionally inhibits the process of angiogenesis. The indirect effect of SSAs on angiogenesis results from their ability to influence the secretion of other pro-angiogenic factors by tumor cells. SSAs can reduce the secretion of factors such as PDGF (platelet-derived growth factor), FGF (fibroblast growth factor), or angiopoietins, which translates into a reduction in the growth of blood vessels. Kumar et al. [213] showed that the reintegration of the SSTR2 gene in pancreatic cancer cells resulted in a significant reduction in the concentration of VEGF and MMP-2, which are key factors of angiogenesis and metastasis. The results suggest that SSTR2 gene transfer may represent a novel gene therapy strategy for the treatment of pancreatic cancer. Similarly, Wang et al. [214] conducted a study to investigate the effects of the somatostatin analogue, octreotide, on invasion and metastasis of gastric cancer in vitro and in vivo. Using a mouse model with SGC-7901 transplanted gastric cancer cells, the researchers observed that administration of octreotide reduced the number of invasive cancer cells and reduced the number of tumor metastases compared to the control group. In addition, a decrease in microvascular density and VEGF and MMP-2 expression was observed in tissues treated with octreotide. The conclusion of the study is that octreotide inhibits the invasion and migration of SGC-7901 gastric cancer cells in vitro and reduces tumor metastasis in vivo by inhibiting MMP-2 expression and reducing tumor angiogenesis.

## 7. Challenges in NET Therapy: Navigating Resistance, Diagnostic Sensitivity, and Long-Term Effectiveness

The therapy of neuroendocrine tumors using somatostatin analogs is undeniably an effective strategy, but it is associated with several significant limitations. One of the main challenges is the potential development of drug resistance, which may be linked to reduced expression of somatostatin receptors on the surface of tumor cells, limiting the effectiveness of treatment [215,216]. Another crucial limitation is the reduced sensitivity of certain diagnostic methods, such as somatostatin receptor scintigraphy (SRS), in certain types of NETs, for example, small bowel neuroendocrine tumors (SBNETs) [217]. This complicates the precise localization of the tumor and the accurate determination of disease extent, which can have significant clinical and therapeutic implications. Additionally, while somatostatin analogs can control symptoms and delay disease progression, their ability to effectively reduce tumor size is limited [218,219]. Only about 30% of patients experience physical tumor shrinkage, emphasizing the need to explore more effective therapies [220,221]. The frequency of drug administration, especially in the case of formulations requiring twice-daily dosing, and the associated inconvenience for patients, can pose additional challenges to effective therapy. Prospective studies involving a substantial number of patients are essential for determining the optimal dosage and administration modes of somatostatin analogs. There is a need for research focused on the development of new, slow-release compounds that are specific to somatostatin receptor subtypes [55]. Furthermore, the costs of treatment can be a significant burden, particularly in healthcare systems where patients are responsible for a portion of the treatment costs [222,223,224]. In the long-term perspective, therapy with somatostatin analogs may encounter limitations in terms of tumor regression and complete cure.

Despite the fact that SSAs can induce sustained disease stabilization in the case of NETs, resistance to treatment often occurs after prolonged use, even with dose intensification. This resistance may result from various mechanisms, often dependent on the clinical context [215,216]. One potential resistance mechanism involves a reduction in the expression of somatostatin receptors on the surface of tumor cells. As treatment duration extends, tumor cells may undergo genetic and epigenetic changes, leading to a decrease in receptor numbers, thereby reducing the effectiveness of somatostatin analogs. Another aspect is the potential occurrence of mutations in the genes encoding somatostatin receptors. These mutations can alter receptor structure, affecting their ability to bind to somatostatin analogs and transmit antitumor signals. Additionally, the activation of alternative signaling pathways independent of somatostatin receptors may contribute to treatment resistance. The heterogeneity of somatostatin receptor expression in different tumor areas is also a crucial aspect of resistance. Regions with lower somatostatin receptor expression may exhibit reduced susceptibility to the action of somatostatin analogs, resulting in limited treatment efficacy in these areas [216,225,226]. Therefore, understanding the mechanisms of resistance to somatostatin analog treatment and their relationship with somatostatin receptor expression is crucial for developing more effective therapeutic strategies in the treatment of neuroendocrine tumors.

During the therapy with somatostatin analogs, the phenomenon of receptor desensitization and tachyphylaxis is observed, which constitutes significant aspects influencing the effectiveness of treatment [227,228]. Receptor desensitization refers to the process in which cells lose their response to constant or prolonged exposure to stimuli, including the presence of somatostatin analogs [117,229]. Tachyphylaxis, on the other hand, is the rapid decline in cellular response to repeated doses of the drug. The mechanisms of somatostatin receptor expression play a crucial role in these phenomena [218,230]. Receptor–ligand interactions, involving the interplay between the ligand and SSRT receptor, are essential for initiating cellular signaling. In the case of desensitization, prolonged exposure to somatostatin analogs can lead to receptor depletion or internalization, reducing the effectiveness of therapy [117,231]. Tachyphylaxis may result from a similar mechanism and influence on signaling pathways, causing a loss of cellular responsiveness [218,230]. Therefore, understanding these mechanisms is crucial for developing therapeutic strategies that minimize these phenomena. Possible approaches include cyclic breaks in therapy, dose adjustments, or the use of a combination of different somatostatin analogs. Research into a better understanding of SSRT expression and the regulation of these mechanisms may contribute to the development of more effective and durable therapies for patients with neuroendocrine diseases.

Somatostatin analogs constitute a crucial therapeutic tool in the treatment of neuroendocrine tumors; however, their usage may be associated with certain side effects. Among the most common adverse effects are fatigue, sleep disturbances, pain and discomfort, nausea, abdominal cramping, diarrhea, steatorrhea, flatulence, hyperglycemia, and biliary sludging [218,220,232]. Understanding the mechanisms of somatostatin receptor expression is key to explaining these side effects. Interactions between the ligand (somatostatin analog) and SSRT receptors can lead to various effects, both therapeutic and undesired. For instance, the inhibition of hormone release by neuroendocrine cells, one of the intended effects of SSAs, may simultaneously contribute to fatigue and sleep disturbances [232]. The impact of SSAs on the gastrointestinal tract, manifesting as abdominal pain, diarrhea, steatorrhea, and flatulence, may result from their interaction with SSRT receptors present in the digestive system [230,232]. Furthermore, hyperglycemia, an elevated blood glucose level, may be a consequence of SSAs affecting metabolic processes [232,233]. Adverse effects associated with the biliary system, such as biliary sludging, may stem from the effects of SSAs on gallbladder contractions and bile flow [234]. Comprehending these mechanisms is crucial for effectively managing side effects and minimizing their impact on patients’ quality of life. Conclusions from previous research suggest that the majority of side effects depend on individual patient reactions and the dosage of the administered therapy. Further studies on the connections between SSRT expression and side effects can contribute to optimizing therapy and minimizing undesirable effects.

## 8. Conclusions and Future Perspectives

The clinical management of patients with NETs has been significantly transformed by the introduction of somatostatin analogue as crucial diagnostic and treatment tools. Although many patients treated with SSA experience symptomatic relief and stabilization of tumor growth for varying periods, complete tumor regression remains uncommon. In cases where tumor regression does occur, potential mechanisms include the antagonism of local growth factor release and effects, possibly involving activation of tyrosine and serine-threonine phosphatases, along with indirect effects through anti-angiogenesis. To further enhance the clinical management of NET patients, ongoing research focuses on the development of novel SSA, innovative drug combination therapies, and chimeric molecules. These advancements hold promise for improving treatment outcomes. Additionally, gaining a comprehensive understanding of the mode of action of SSA will contribute significantly to refining their effectiveness in managing patients with NETs.

Despite significant progress in the treatment of neuroendocrine tumors using systemic therapy, such as somatostatin analogs or other methods, a clear demonstration of a substantial extension of overall patient survival has not been unequivocally proven. This phenomenon may arise from the biological diversity of neuroendocrine tumors and the complexity of mechanisms regulating their growth and development. In the context of somatostatin receptor expression, various known and putative mechanisms exist that can impact the effectiveness of treatment. Variability in somatostatin receptor expression on the surface of tumor cells can lead to uneven responses to therapy, affecting its overall efficacy. It is possible that there are yet unidentified signaling pathways or alternative receptor mechanisms that influence the outcomes of systemic treatment in NETs. Additionally, understanding the mechanisms of treatment resistance is crucial. This may include mutations in genes encoding somatostatin receptors, leading to changes in receptor structure and affecting their ability to bind to somatostatin analogs. Activation of signaling pathways independent of somatostatin receptors may also impact the effectiveness of systemic therapy. Therefore, further research into the mechanisms of somatostatin receptor expression in NETs and the identification of potential therapeutic targets are necessary to refine systemic treatment strategies and strive for improvement in overall survival for patients with neuroendocrine tumors.

Contemporary preclinical and clinical studies on neuroendocrine tumor therapy, particularly using somatostatin analogs, yield promising results, suggesting potential benefits in symptom control and patient survival. However, to obtain conclusive, binding evidence of the effectiveness of these therapies, further extensive clinical trials, especially of a randomized nature, are necessary. A crucial aspect requiring in-depth analysis is the understanding of SSRT expression mechanisms and their role in therapy response. Receptor–ligand interactions play a key role in the efficacy of somatostatin analogs. Specific connections between the ligand (somatostatin analog) and the SSRT initiate signaling cascades that influence the functions of tumor cells, including their growth and viability. Known mechanisms based on somatostatin receptors include inhibiting adenylate cyclase activity, leading to reduced cyclic AMP (cAMP) production, and modulating MAPK and PI3K/Akt pathways, influencing cell proliferation and survival. Resistance to therapy may result from various factors, and mutations in genes encoding SSRT may be significant, leading to structural changes in receptors. Additionally, the activation of alternative signaling pathways independent of somatostatin receptors can compensate for therapeutic effects, impacting the ultimate evidence of therapeutic efficacy. Therefore, further studies should focus on identifying new SSRT expression mechanisms and their role in therapeutic effectiveness. Integrating this detailed information into the design of clinical trials will enable a fuller understanding of the impact of somatostatin analogs on molecular and clinical mechanisms, ultimately contributing to a better assessment of their therapeutic value in neuroendocrine tumor treatment.

## Figures and Tables

**Figure 1 cancers-16-00116-f001:**
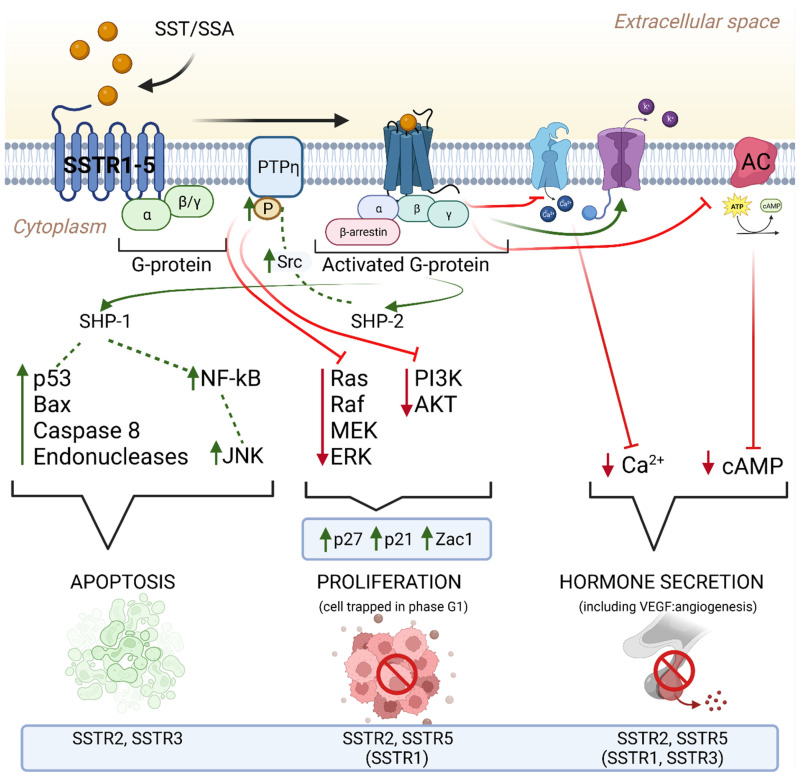
Mechanism of the antitumor effect of SST receptors after ligand binding. This mechanism involves several key signaling pathways: (1) *Adenylate cyclase pathway*: SSTR binds to adenylate cyclase and reduces levels of cyclic adenosine monophosphate (cAMP) in the cell. Reducing the concentration of cAMP leads to the inhibition of protein kinase activity, which in turn prevents the activation of oncogenes and inhibits the development and progression of the tumor. (2) *Tyrosine protein pathway*: SSTR binds to SST, leading to an increase in protein tyrosine phosphatase (PTP), which dephosphorylates and inactivates tyrosine kinase. Many protein kinases can be inhibited, such as mitogen-activated protein kinase (MAPK), leading to inhibition of DNA and protein synthesis. (3) *Phosphatidylinositol 3 kinase (PI3K) pathway*: SSTR increases the expression of p21 and p27 via PI3K, resulting in the inhibition of phosphorylation of PRb and the cyclin E-dependent kinase 2 complex. (4) *Calcium signaling pathway*: SSTR causes an ion exchange between Ca^2+^ and H^+^, which causes a decrease in the concentration of intracellular calcium, an increase in acidification of the intracellular environment and inhibition of cell proliferation.

**Table 3 cancers-16-00116-t003:** Somatostatin analogues and their affinities for somatostatin receptors.

Somatostatin Analogue	Affinity for Somatostatin Receptors	Selectivity	Duration of Action	Application in NET
*Octreotide*	SSTR2 > SSTR5 > SSTR3 > SSTR1 > SSTR4	high	Long	Symptom control, growth inhibition, and hormone control in pancreatic, intestinal, lung, neck, and thyroid NET
*Lanreotide*	SSTR2 = SSTR5 > SSTR3 > SSTR1 > SSTR4	high	Long	Symptom control, growth inhibition, and hormone control in pancreatic, intestinal, lung, and neck NET
*Pasireotide*	SSTR5 > SSTR2 > SSTR3 > SSTR1 > SSTR4	high	Long	Symptom control, growth inhibition, and hormone control in pancreatic and intestinal NET
*Vapreotide*	SSTR2 > SSTR5 > SSTR3 > SSTR1 > SSTR4	high	Short	Diagnosis and assessment of somatostatin receptors in NETs
*Somatuline*	SSTR2 > SSTR5 > SSTR3 > SSTR1 > SSTR4	high	Medium	Symptom control, growth inhibition, and hormone control in pancreatic, intestinal, lung, neck, and thyroid NET

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
