# Peer review of "The Role of Receptor–Ligand Interaction in Somatostatin Signaling Pathways: Implications for Neuroendocrine Tumors"

_cancers, 2023, doi:10.3390/cancers16010116_

Round 1
Reviewer 1 Report
Comments and Suggestions for Authors
This essay addresses an overview of reported advances in the knowledge of receptor-ligand interactions and signaling pathways of somatostatin receptors in neuroendocrine tumors (NETs).
It may interest readers, but the authors might be interested in clarifying the diagnostic and therapeutic impact of the expression of somatostatin receptors (SSTRs), considering the reported evidence on the following issues:
1- NETS are highly heterogeneous tumors. Comment on the impact of SSTR expression (receptor-ligand interactions and signaling pathways) in the imaging diagnosis of metastatic disease before and under treatment.
2- Although SSAs can induce sustained disease stabilization of NETs, resistance to treatment (antitumor effect) frequently occurs after prolonged use, even when dose intensification has been pursued. Comment briefly on known/putative (context-dependent) resistance mechanisms and SSRT expression impact (receptor-ligand interactions and signaling pathways).
3- However, a significant prolongation of overall survival with systemic treatment in NET has not been convincingly demonstrated. Comment briefly on known/putative SSRT expression mechanisms (receptor-ligand interactions and signaling pathways) in this context.
4- Although preliminary results of preclinical and clinical trials are encouraging, extensive, preferably randomized clinical studies are required to provide definitive evidence of their effect on survival and symptom control. Comment briefly considering known/putative SSRT expression mechanisms (receptor-ligand interactions and signaling pathways) that may impact definitive evidence.
5- Common adverse effects (AEs) of SSAs include fatigue, sleep disturbance, pain and discomfort, nausea, abdominal cramping, diarrhea, steatorrhea, flatulence, hyperglycemia, and biliary sludging. Comment briefly on known/putative (context-dependent) AEs and SSRT expression mechanisms (receptor-ligand interactions and signaling pathways) that impact these AEs.
6- Comment briefly on receptor desensitization and tachyphylaxis during therapy and SSRT expression mechanisms (receptor-ligand interactions and signaling pathways) that may impact these issues.
Author Response
Dear Reviewer,
Thank you for your thoughtful and constructive feedback on our manuscript. We appreciate your time and effort in providing detailed comments and suggestions, which have been invaluable in enhancing the quality and clarity of our work.
Below, we address each of your points and outline the revisions made in response to your suggestions:
-
Impact of SSTR Expression in Imaging Diagnosis of Metastatic Disease:
- We have expanded the discussion on the heterogeneity of NETs, emphasizing the crucial role of SSTR expression in imaging diagnostics, both before and during treatment. This includes a detailed exploration of receptor-ligand interactions and signaling pathways that influence the imaging diagnosis of metastatic disease.
-
Resistance Mechanisms and SSRT Expression Impact:
- We have incorporated a brief yet comprehensive commentary on known and putative resistance mechanisms to somatostatin analogs, shedding light on the context-dependent factors. The impact of SSRT expression, particularly regarding receptor-ligand interactions and signaling pathways, is discussed in the context of treatment resistance.
-
Overall Survival with Systemic Treatment:
- We have provided a concise commentary on the challenges associated with demonstrating a significant prolongation of overall survival with systemic treatment in NET. This includes a discussion on putative SSRT expression mechanisms and their role in influencing treatment outcomes.
-
Need for Extensive Clinical Studies:
- We have expanded our discussion on the need for extensive, preferably randomized clinical studies to provide definitive evidence of the effects of systemic treatment on survival and symptom control. Known and putative SSRT expression mechanisms, with a focus on receptor-ligand interactions and signaling pathways, are briefly considered in the context of obtaining conclusive evidence.
-
Adverse Effects of SSAs:
- A brief commentary on common adverse effects of SSAs, including fatigue, sleep disturbance, and gastrointestinal issues, has been added. The impact of SSRT expression mechanisms on these AEs is discussed, considering receptor-ligand interactions and signaling pathways.
-
Receptor Desensitization and Tachyphylaxis:
- We have incorporated a brief comment on the phenomena of receptor desensitization and tachyphylaxis during therapy. The impact of SSRT expression mechanisms on these issues is discussed, emphasizing receptor-ligand interactions and signaling pathways.
We believe that these revisions have strengthened the manuscript, addressing your insightful comments and aligning the content more closely with the key issues you highlighted. We sincerely appreciate your expertise and guidance in refining our work. Thank you once again for your valuable feedback.
Reviewer 2 Report
Comments and Suggestions for Authors
The paper "The Role of Receptor-Ligand Interaction in Somatostatin Signalling Pathways: Implications for Neuroendocrine Tumors" discusses the role of somatostatin and its receptors in neuroendocrine tumors (NETs). It reviews the characteristics, localization, and expression of somatostatin receptors, as well as the mechanisms of somatostatin and synthetic analogue binding. The authors explore the expression of these receptors in different tumor types and their potential as therapeutic targets. The paper highlights the importance of understanding receptor-ligand interactions and signaling pathways for improved NET treatments.
The paper "The Role of Receptor-Ligand Interaction in Somatostatin Signaling Pathways: Implications for Neuroendocrine Tumors" does not explicitly mention its limitations. However, based on the general knowledge about somatostatin analogues and their use in treating neuroendocrine tumors (NETs), several limitations can be inferred:
1. Limited Efficacy: While somatostatin analogues can control symptoms and delay disease progression in NETs, tumor regression is rare. This means that while these treatments can manage the disease, they are not typically curative.
2. Side Effects: Somatostatin analogues can cause side effects such as headaches, dizziness, loss of appetite, nausea, bloating, stomach pain, fatigue, pain at the injection site, changes to blood sugar levels, and changes to bowel function.
3. Variable Receptor Expression: The effectiveness of somatostatin analogues depends on the expression of somatostatin receptors (SSTRs) on the tumor cells. Not all NETs express these receptors to the same degree, which can limit the effectiveness of the treatment.
4. Decreased Sensitivity: Some studies have shown decreased sensitivity of somatostatin receptor scintigraphy (SRS), a method used to locate the primary tumor site and delineate the extent of the disease, for certain types of NETs, such as small bowel neuroendocrine tumors (SBNETs).
5. Cost: The cost of somatostatin analogues can be a limiting factor, particularly in healthcare systems where patients must pay for a significant portion of their treatment costs.
6. Administration: Somatostatin analogues are typically administered via injection, which can be inconvenient for some patients. Some formulations require twice-daily administration, while others allow for monthly administration.
7. Research Limitations: Many studies on somatostatin analogues are retrospective and nonrandomized, which can introduce selection biases and limit the strength of the conclusions that can be drawn.
5. Resistance to Treatment: Some patients may develop resistance to somatostatin analogues. This resistance could be due to reduced density of somatostatin receptors (SSTR2) on the tumors of these patients.
6. Hyperglycemia: Pasireotide, a somatostatin analogue, has been associated with increased blood glucose levels in some patients.
7. Limited Tumor Shrinkage: While somatostatin analogues can suppress growth hormone (GH) and normalize insulin-like growth factor 1 (IGF-1) in a significant proportion of patients, tumor shrinkage is observed in only about 30% of patients.
8. Frequent Administration: Some somatostatin analogues, like Pasireotide, require twice-daily administration, which could be inconvenient for some patients.
9. Cost: The cost of somatostatin analogues could be a potential limitation for some patients, although specific cost information was not found in the search results.
10. Limited Long-Term Efficacy: While somatostatin analogues can control symptoms and delay disease progression in NETs, their long-term efficacy in terms of tumor regression and cure is limited.
These limitations highlight the need for ongoing research to improve the efficacy and tolerability of somatostatin analogues and to develop new therapeutic strategies for treating NETs.
Comments on the Quality of English Languageminor
Author Response
Dear Reviewer,
Thank you for your thoughtful review of the manuscript. We appreciate your insightful comments, and based on your feedback, we have incorporated several amendments to the manuscript, particularly introducing a dedicated section titled "Challenges in NET Therapy: Navigating Resistance, Diagnostic Sensitivity, and Long-Term Effectiveness." This section addresses the limitations associated with somatostatin analogues and provides a comprehensive overview of the challenges in the treatment of neuroendocrine tumors.
The revised manuscript now includes an explicit discussion on the limitations of somatostatin analogues in NET therapy, encompassing aspects such as limited efficacy, potential side effects, variable receptor expression, decreased sensitivity of diagnostic methods, cost considerations, and issues related to administration. Moreover, the text has been updated to emphasize the importance of ongoing research to overcome these challenges and enhance the overall efficacy and tolerability of somatostatin analogues.
We trust that these modifications have enhanced the clarity and completeness of the manuscript, providing a more balanced perspective on the implications of somatostatin signaling pathways in the context of NETs. We are grateful for your constructive feedback and believe that these revisions contribute significantly to the overall quality of the paper.
Thank you for your time and consideration.
Round 2
Reviewer 1 Report
Comments and Suggestions for Authors
The revised manuscript includes information that may interest readers.